# Quantifying Uncertainty of Uplift

## Abstract

Uplift modeling refers to the task of estimating the causal effect of a treatment on an individual, also known as the conditional average treatment effect. Despite significant progress in uplift methods in recent years, the uncertainty of the estimates has been largely ignored in the literature. We explain why estimating uncertainty of the treatment effect is particularly important in many common use cases and we define epistemic uncertainty of the uplift estimates. Our main goal is to explain how uncertainty estimates can be incorporated into commonly used uplift model families with relative ease, and to demonstrate this we provide details for two practical methods that build on the standard approaches but add support for uncertainty quantification. We illustrate the methods on three real datasets and show how information about the uncertainty can be used in uplift modeling tasks, and additionally quantify the accuracy of the uncertainty estimates on simulated data.

## 1 Introduction

Uplift modeling – also known as heterogeneous treatment effect estimation – is a form of causal inference providing answers to causal questions like "Will this medicine make you better?" (Kamath et al., 2022; Falet et al., 2022; Alaa & Schaar, 2017), "Which students need an intervention not to drop out?" (Olaya et al., 2020), and "Is it profitable to offer a discount to this customer?" (Haupt & Lessmann, 2022). In essence, the goal is to estimate the effect of a potential treatment on *individual observations* (persons in all of the examples above) and our interest here is in cases of binary outcomes. The magnitude of the effect is also called the Conditional Average Treatment Effect (CATE) (Rubin, 1974).

Uplift modeling has received increasing attention during the past years with advances in e.g. cost-sensitive uplift modeling (Verbeke et al., 2022), techniques for addressing imbalance in the treatment group sizes (Xu & Yadlowsky, 2022; Zhong et al., 2022), and handling of high class imbalance (Nyberg & Klami, 2023). Somewhat surprisingly, the question of *uncertainty* of the uplift estimates has remained largely understudied until very recently. The focus in core uplift literature has been on providing point estimates for the treatment effects with no attention paid to their variation, with a few papers addressing the uncertainty starting to appear only recently.

Bokelmann & Lessmann (2022) and Gutierrez & Gérardy (2017) studied the variation of *uplift metrics* over the whole population but did not address the uncertainty for individual treatment effects. The need for quantifying uncertainty of the estimates at the individual level was pointed out already by Hill (2011), but they studied the question only in the context of continuous outcomes where Bayesian treatment of regression is sufficient and proposed a specific method suitable only for that setting. Hahn et al. (2020) extends their work and discusses uncertainty in detail, but is still limited to continuous outcomes. Huang et al. (2022) noted that the uncertainties in these contexts should be estimated but settled for quantifying the uncertainties of the two regression models in the T-model, rather than the uplift itself. Alaa & Schaar (2017) demonstrated that measuring individual variation can be useful in the context of heart transplant prioritization and quantified it based on the uncertainty of Gaussian Process regression for the different treatment groups, but again they did not consider the variation in the uplift estimate itself. Also these focused on continuous outcomes.

Another stream of research focusing quantification of uncertainty has been presented in the context of estimating CATE in presence of latent confounders (Kallus et al., 2019; Louizos et al., 2017; Jesson et al.,

2021; Lei & Candès, 2021). The confounders bias the CATE estimates themselves and these works provide interval estimates with proven guarantees for capturing the true CATE but all build on somewhat elaborate technical solutions and are not easy to extend for other modeling choices. Kallus et al. (2019) develops a specific kernel-regression method, Louizos et al. (2017); Jesson et al. (2020) build on causal extension of variational autoencoders, Jesson et al. (2021) uses a combination of Bayesian neural networks and mixture models, and Lei & Candès (2021) considers only methods framed using the conformal inference principle. In summary, these recent works all make the same observation as we do regarding importance of quantifying the uncertainty of CATE estimates and provide methods for estimating it. However, it remains difficult for practitioners to account for uncertainty within the more classical uplift estimation methods, for instances in cases where they are already using an instance of T-learners (Soltys et al., 2015; Künzel et al., 2019) of decision trees (Athey & Imbens, 2016) for estimating the effects. Our main goal is to show how these common model families can be extended to account for uncertainty.

A considerable part of all uplift research has been in the context of advertising or marketing (Haupt & Lessmann, 2022; Ke et al., 2021; Zhao et al., 2022; Gu et al., 2021; Moriwaki et al., 2021). This may partially explain the lack of interest in quantifying the uncertainty. In such cases reliable estimates for the *average* effect is usually sufficient for financial gains. However, even in these applications properly quantifying the uncertainty of the estimates can be important, as we will empirically show in this work by highlighting that the confidence intervals for the uplift estimates for commonly used benchmark data sets are extremely wide. Especially in the exploration phase when deciding whether to apply a treatment to an observation for optimal data collection, uncertainty is essential. Accurate quantification of the uncertainty is also likely to be useful from the perspective of the marketer to better understand the population and the models. For instance, if the uncertainty is high for all observations, it implies that the model has not learned much from the data.

In many other uses cases, in particular in the medical domain, adequately assessing the uncertainty becomes a necessity. Recently Falet et al. (2022) investigated the use of uplift modeling to target treatment of a medication for multiple sclerosis (MS) showing clearly how identifying subgroups that benefit from a medication can help save lives. However, to progress from analysis of effects towards practical medical recommendations requires knowing also the (potentially high) uncertainty of the estimates to be used as additional information for making the subjective decision. For instance, in a case of terminal illness a patient may prefer a treatment with high uncertainty to have a chance of additional months or years of life-time even when the expected effect is slightly negative. In other cases we may want to ensure that a medication is not given to any individual that reacts negatively.

To address these needs, we define uncertainty of uplift estimates and present two methods of estimating it in the context of two broadly used uplift modeling families. We show how the double-classifier (or T-learner (Künzel et al., 2019)) can be modified to quantify the uncertainty by using well-calibrated probabilistic base classifiers, here the Dirichlet-based Gaussian Process (DGP) by Milios et al. (2018). We also show how tree-based methods can be modified to provide uncertainty estimates and extend the honest tree by Athey & Imbens (2016) as a practical example. We demonstrate the behavior of both methods on three commonly used benchmark datasets focusing in particular on illustrating the benefits of explicitly quantifying the uncertainty in this scenario and characterising how it depends on the amount of available training data and the chosen method. The main conclusion of the empirical experiments is that for commonly used uplift datasets the uncertainty of the estimates is large and cannot be ignored in practical applications. Finally, we evaluate on semi-synthetic data with known treatment effects how well the uncertainty estimates are calibrated.

## 2 Problem Formulation

Throughout the paper we use notational convention where scalar random variables are indicated by standard serif fonts, e.g. u, and their realizations by italic $u$. Vector-valued random variables and realizations are indicated by boldface $\mathbf{x}$ and $\boldsymbol{x}$ respectively. Finally, we use u($\boldsymbol{x}$) to denote u$|\mathbf{x} = \boldsymbol{x}$ for conditioning the random variable on having observed $\mathbf{x}$ to take value $\boldsymbol{x}$.

In this section we formulate the problem of estimating uncertainty of uplift estimates. Uplift $\tau(\boldsymbol{x})$ for an individual characterized by some features $\boldsymbol{x}$ is classically defined as

$$\tau(\boldsymbol{x}) = P(\mathrm{y} = 1|\boldsymbol{x}, \mathrm{do}(w = 1)) - P(\mathrm{y} = 1|\boldsymbol{x}, \mathrm{do}(w = 0)) \qquad (1)$$

where $P(\mathrm{y} = 1|\boldsymbol{x}, \mathrm{do}(w = 1))$ is the conditional probability for a positive outcome $\mathrm{y} = 1$ if a treatment (denoted by $w = 1$) is applied, and $P(\mathrm{y} = 1|\boldsymbol{x}, \mathrm{do}(w = 0))$ is the corresponding probability if a treatment is *not* applied. Further, do() is the do-operator (Pearl, 2009). When the data is collected in a randomized controlled trial, the requirements of the do-operator are satisfied and the notation simplifies to conditioning on $w$. We drop the do-notation in this paper for clarity of presentation, although note that the equations are valid also without this assumption given that the requirements are satisfied in some other way, e.g. by inverse probability of treatment weighting (IPTW Austin, 2011)

Sometimes the uplift $\tau(\boldsymbol{x})$ is called Conditional Average Treatment Effect (CATE) (Rubin, 1974; Gutierrez & Gérardy, 2017) which emphasizes its characteristic properties. Here *conditional* refers to estimating the effect conditionally on $\boldsymbol{x}$ (the properties of the individual) whereas *average* refers to the expected effect. We seek to characterize the *epistemic uncertainty of CATE.* That is, we want to quantify the uncertainty of $\tau(\boldsymbol{x})$ stemming from needing to estimate $P(\mathrm{y} = 1|\boldsymbol{x}, w)$ from finite data. We do this by re-defining the uplift estimate itself as a random variable

$$\mathrm{u}(\boldsymbol{x}) = \mathrm{t}(\boldsymbol{x}) - \mathrm{c}(\boldsymbol{x}) \qquad (2)$$

where $\mathrm{u}(\boldsymbol{x})$ is the uplift, and $\mathrm{t}(\boldsymbol{x})$ and $\mathrm{c}(\boldsymbol{x})$ refer to the unknown probabilities for $P(\mathrm{y} = 1|\boldsymbol{x}, w)$ for $w = 1$ and $w = 0$, respectively. For convenience of notation that avoids explicit reference to $w$, we use t for "treatment" and c for "control", following the convention where the untreated group is called the control group. The definition naturally extends equation 1 in the sense that the expectation of equation 2 matches the classical definition: $\mathbb{E}[\mathrm{u}(\boldsymbol{x})] = \tau(\boldsymbol{x})$. The support of $\mathrm{u}(\boldsymbol{x})$ is $[-1, 1]$ with the end point of 1 corresponding to the perfect treatment effect (i.e. an individual with zero probability of positive outcome without treatment is guaranteed to have one after a treatment).

The probability density function (PDF) of $\mathrm{u}(\boldsymbol{x})$ is

$$p(\mathrm{u}(\boldsymbol{x}) = u) = \begin{cases} \int_u^1 p(\mathrm{t}(\boldsymbol{x}) = t, \mathrm{c}(\boldsymbol{x}) = t - u)dt, & u \geq 0 \\ \int_0^{1+u} p(\mathrm{t}(\boldsymbol{x}) = t, \mathrm{c}(\boldsymbol{x}) = t - u)dt, & u < 0 \end{cases} \qquad (3)$$

where the boundaries of the two integrals cover all values of $t$ and $c$ that have support. E.g. for $\mathrm{u}(\boldsymbol{x}) = 0$ the integral is from 0 to 1 as $\mathrm{u}(\boldsymbol{x}) = 0$ when $\mathrm{t}(\boldsymbol{x}) = \mathrm{c}(\boldsymbol{x})$ and both $\mathrm{t}(\boldsymbol{x})$ and $\mathrm{c}(\boldsymbol{x})$ only take values in $[0, 1]$. Further, the corresponding cumulative distribution function (CDF) is

$$\int_{-1}^u p(\mathrm{u}(\boldsymbol{x}) = u)du = \begin{cases} \int_0^1 \int_{t+u}^1 p(\mathrm{t}(\boldsymbol{x}) = t, \mathrm{c}(\boldsymbol{x}) = c)dcdt, & u \geq 0 \\ \int_0^{1-u} \int_v^1 p(\mathrm{t}(\boldsymbol{x}) = t, \mathrm{c}(\boldsymbol{x}) = c)dcdt, & u < 0 \end{cases} \qquad (4)$$

where $v = \max(0, t + u)$. To estimate the density of $\mathrm{u}(\boldsymbol{x})$ we hence need to estimate the joint density $p(\mathrm{t}(\boldsymbol{x}), \mathrm{u}(\boldsymbol{x})) = p(\mathrm{t}, \mathrm{u}|\mathbf{x} = \boldsymbol{x})$. We assume that conditional on $\mathbf{x} = \boldsymbol{x}$ the two are independent, so that the joint density factorizes and it is sufficient to estimate the distributions for $\mathrm{t}(\boldsymbol{x})$ and $\mathrm{c}(\boldsymbol{x})$ separately. This is the same simplifying assumption many practical CATE models already do. Next we explain how this can be done in practice in context of commonly used model families.

## 3   Methods

Even though the formal definition of $\mathrm{u}(\boldsymbol{x})$ in equation 2 is straightforward, the practical process of estimating the density requires some care. In this work we present the details for two example models from two commonly used uplift model families. The first case describes a general process building on the widely adopted double-classifier or T-learner (Soltys et al., 2015; Künzel et al., 2019) and presents a practical method that uses well-calibrated Gaussian Process classifiers by Milios et al. (2018) as base classifiers, and the second case is a novel tree-based model extending the honest tree by Athey & Imbens (2016). Both methods are presented as demonstrations of how uncertainty quantification can be incorporated to commonly used model families

with only minor additions; for T-learner it is sufficient to use base learners that quantify uncertainty, whereas for tree-based models we get uncertainty estimates by standard conjugate analysis.

For both approaches the basic idea is to represent the distribution of $u(\boldsymbol{x})$ explicitly as the difference between empirical estimates of the distributions of $t(\boldsymbol{x})$ and $c(\boldsymbol{x})$. For the tree-based model we will have a closed-form solution as the Beta-difference distribution (Pham-Gia & Turkkan, 1993), whereas for the double classifier we will use a Monte Carlo approximation to characterise the distribution. This allows easy visualization of the uncertainty as well as numeric computation of e.g. expectations (matching the usual definition of equation 1) or highest posterior density (HPD) intervals (Chen & Shao, 1999).

### 3.1 Double Classifier with Dirichlet-Based Gaussian Processes

The double classifier approach for classical uplift modeling estimates both $P(y = 1|\boldsymbol{x}, w = 1)$ and $P(y = 1|\boldsymbol{x}, w = 0)$ separately based on the treatment and control subsets with a suitable classifier and computes the uplift as their difference. Despite its simplicity, this approach remains one of the more competitive approaches; see Olaya et al. (2020) and Nyberg & Klami (2023) for recent comparisons.

The double classifier builds fundamentally on the assumption that the two probabilities are independent, and we retain this assumption. Then this approach generalizes directly to estimating the density of $u(\boldsymbol{x})$ as long as the classifiers provide densities characterising $t(\boldsymbol{x})$ and $c(\boldsymbol{x})$. In principle any such classifier would work, but in practice we want classifiers that provide *well-calibrated* estimates. A well-calibrated estimate refers to one that accurately characterises the distribution and does not e.g. over- or underestimate the uncertainty. We use the Dicirhlet-based Gaussian Process (DGP) model by Milios et al. (2018) that has been demonstrated to provide well-calibrated estimates in a range of classification tasks, but e.g. other GP-based classifiers would also be applicable.

A DGP for a binary classification problem (either for $t(\boldsymbol{x})$ or for $c(\boldsymbol{x})$) is constructed using two latent functions $f_1(\boldsymbol{x})$ and $f_0(\boldsymbol{x})$, one for each class ($y = 1$ and $y = 0$). Both functions follow the same GP prior with a suitably chosen kernel function and two latent functions are used to support heteroscedastic noise required for improving the calibration over standard GP classifiers. The gist of the model is that the latent functions parameterise the shapes of class-specific gamma distributions which can be normalized over the classes to form a Dirichlet distribution over the class probabilities (two in this case). In practice the algorithm is made computationally efficient by approximating the gamma distributions with lognormal-distributions for suitably transformed shape parameters. A detailed derivation is provided by Milios et al. (2018), but below we present the final approximation in our notation.

If we denote by $\alpha_\epsilon$ a prior parameter for the assumed Dirichlet, then the model transforms the positive labels ($y = 1$) into $\hat{y} = \log(1 + \alpha_\epsilon) - \frac{1}{2}\log(1/(1 + \alpha_\epsilon) + 1)$ and the negative labels into $\hat{y} = \log \alpha_\epsilon - \frac{1}{2}\log(1/\alpha_\epsilon + 1)$. Given this transformation, we obtain calibrated class probabilities by combining the GP priors with the likelihood

$$p(\hat{y}|f_1) = \mathcal{N}\left(f_1, \log(1/(1 + \alpha_\epsilon) + 1)\right) \tag{5}$$

for the positive labels and

$$p(\hat{y}|f_0) = \mathcal{N}\left(f_0, \log(1/\alpha_\epsilon + 1)\right) \tag{6}$$

for the negative ones. Since the likelihood is normal, we can use exact inference for computing the posterior over the latent functions, making the approach highly robust and easy to use. We do this in the experiments to avoid contaminating the results with potentially hard-to-interpret approximation errors, but Milios et al. (2018) explains how the method trivially scales for larger datasets by using standard sparse variational approximations (Titsias, 2009).

We apply DGP for estimating $u(\boldsymbol{x})$ by learning separate DGP models for the treatment and control groups. It is easy to sample from the DGP predictive distribution for any $\boldsymbol{x}$ by sampling $\hat{y}$ from the normal marginals and hence we can construct observations of $u(\boldsymbol{x})$ by computing the difference between observations from $t(\boldsymbol{x})$ and $u(\boldsymbol{x})$. An important observation is that if the estimates for $t(\boldsymbol{x})$ and $c(\boldsymbol{x})$ are well-calibrated then so is the estimate for $u(\boldsymbol{x})$, and due to linearity any possible error will at most double.

To apply the model, we need to select the kernel hyperparameters (length scale and noise level) which is done using standard marginal likelihood maximization. This leaves $\alpha_\epsilon$ as the only additional parameter

and for simplicity we use a common $\alpha_\epsilon$ for both $t(\boldsymbol{x})$ and $c(\boldsymbol{x})$ chosen to maximize the joint training data log-likelihood of these classifiers. Milios et al. (2018) showed that $\alpha_\epsilon$ maximizing the log-likelihood of the training data results in well-calibrated classifiers, which is exactly what is needed for calibrated estimation of $u(\boldsymbol{x})$.

### 3.2 Honest Uplift Tree

Uplift trees and uplift (random) forests are popular uplift models (Friedberg et al., 2020; Athey et al., 2019; Oprescu et al., 2019) and hence used here as another example family. Since trees provides an explicit partitioning $\Pi = \{\ell_1, \ldots, \ell_M\}$ of the feature space into leaves $\ell_m$, they provide a natural way of estimating the uplift $\tau(\boldsymbol{x})$ for each leaf $m$ based on the observations of both treatment groups falling into the leaf. We denote by $N_{m,y,w}$ the number of training data observations in the set $\{\boldsymbol{x} \in \ell_m, y, w\}$, so that e.g. $N_{m,y=1,w=1}$ counts the treated observations with positive outcome in the $m$th leaf. The classical uplift estimate is then computed as

$$\tau(\boldsymbol{x}) = \frac{N_{m,y=1,w=1}}{N_{m,y=1,w=1} + N_{m,y=0,w=1}} - \frac{N_{m,y=1,w=0}}{N_{m,y=1,w=0} + N_{m,y=0,w=1}} \tag{7}$$

for all $\boldsymbol{x} \in \ell_m$. In contrast to double classifiers, we only need a single model that is trained on all data, but naturally the conditioning variable $w$ needs to be accounted for in the training process to ensure that all leaves have sufficiently many instances from both groups to obtain reliable estimates.

Next we explain how we can use any uplift tree for estimating the distribution in equation 2 of the uplift estimates. By the nature of a tree, the probabilities $P(y = 1|\boldsymbol{x}, w = 1)$ and $P(y = 1|\boldsymbol{x}, w = 0)$ within a leaf are assumed to be constant and hence also independent. Since all observations are binary, we make the natural assumption that they follow a Bernoulli distribution with unknown parameters $p_t$ and $p_c$ with conjugate beta priors for both rates. The corresponding posterior distributions for each leaf $m$ are then

$$\begin{aligned} t_m(\boldsymbol{x}) &\sim \mathrm{Beta}(\alpha_0 + N_{m,y=1,w=1}, \beta_0 + N_{m,y=0,w=1}), \\ c_m(\boldsymbol{x}) &\sim \mathrm{Beta}(\alpha_0 + N_{m,y=1,w=0}, \beta_0 + N_{m,y=0,w=0}), \end{aligned} \tag{8}$$

where $\alpha_0$ and $\beta_0$ are the parameters of the prior. We use $\alpha_0 = \beta_0 = 1$ corresponding to a uniform prior in our experiments, but additional prior information about e.g. base treatment effect rates could also be encoded here.

Since both $t_m(\boldsymbol{x})$ and $c_m(\boldsymbol{x})$ are Beta distributions, the distribution of $u_m(\boldsymbol{x})$ for each leaf follows a Beta-difference distribution (Pham-Gia & Turkkan, 1993). Even though it is not a commonly used distribution, there are analytic expressions for its moments and there are known algorithms for computing certain conditional probabilities exactly (Raineri et al., 2014). However, for the HPD-interval that we use to evaluate the methods, we still need numerical computation and use the same Monte Carlo approach as for the double classifier.

**Training** The equations above hold for any tree (or, in fact, for any uplift model that partitions the data into disjoint sets of samples). A tree that is good for estimating equation 7 can be trained based on several different criteria.

Following Athey & Imbens (2016), we perform a variable transformation and create a new variable $z$ so that $z_i = 1$ when $y_i = w_i$. Otherwise $z_i = 0$. For this transformation we have $\mathbb{E}[z|\boldsymbol{x}] = \tau(\boldsymbol{x})$ and hence a tree that accurately predicts $z$ is considered an uplift model. We construct a standard CART tree (Breiman et al., 1984) for this so that we have two parameters controlling the complexity of the three: the maximum number of leaf nodes $M_{\max}$ and the minimum number of observations per node $N_{\min}$. Note that even though the tree is trained using $z$, the actual uplift estimates are computed based on the counts.

**Reliable estimates** The *honest tree* proposed by Athey & Imbens (2016) ensures that the estimated uplifts are unbiased by using a separate calibration set for estimating the counts for equation 7. Instead of using $N$ computed from the training data, they use $\hat{N}$ computed from calibration data that is disjoint from the observations used for learning the tree. Even though needing to use a separate calibration set reduces the data efficiency of the model, we prefer this approach as it also means that we do not have to make the

additional assumption of the the leaves producing unbiased estimates from the training data. We believe that the advantage of increased trust in the estimates is essential. Following this idea, we use $\hat{N}$ estimated from a separate calibration set for computing equation 8. This is likely to improve the reliability of the uncertainty estimates and in our preliminary experiments seemed to improve the overall performance.

**Data imbalance**  Recently Nyberg & Klami (2023) showed that tree-based models perform poorly in cases where the proportion of positive outcomes is very small and suggested the use of undersampling to mitigate this. This issues is likely to be even more severe when attempting to model the uncertainty, and hence we incorporate their *stratified undersampling* mechanism.

Stratified undersampling is done by dropping negative observations so that $P^*(y = 1|w) = k \cdot P(y = 1|w)$ where $P(y = 1|w)$ is estimated from data and $P^*(y = 1|w)$ is the resulting positive rate in the data *after undersampling*. This is done separately for the subsets of the data with $w = 1$ and $w = 0$ but with one common factor $k$ selected by cross validation (here 5-fold) to maximize the uplift performance metric AUUC (explained in Section 4.1). Note that we only undersample the training set, not the calibration set, and hence the estimates used for equation 8 correspond to the correct quantities. In the experiments we only use undersampling for the datasets with high class imbalance.

## 4 Experiments

We demonstrate and evaluate the methods using three common uplift benchmark datasets described in Section 4.1. We first demonstrate the new opportunities and insights revealed by explicit investigation of the uncertainty of the estimates in Section 4.2 and then proceed to assess the methods in a more quantitative manner in Section 4.3. The code for reproducing the experiments is provided in the Supplement and will be made available upon publication of the paper.

### 4.1 Data, Model Details and Metrics

**Data.**  We work with three publicly available uplift datasets: `Criteo` (Diemert et al., 2018), `Starbucks` (Rößler et al., 2021), and `Hillstrom` (Radcliffe, 2008).

`Criteo` is the largest publicly available data with 13,979,592 observations from an advertising context. We use the *conversion* label with high class imbalance – only 0.292% of the observations have a positive outcome. `Starbucks` is an e-commerce dataset with 126,184 observations. Finally, `Hillstrom` is a classic uplift dataset also from the e-commerce sector. We used the *Mens E-Mail* as treatment ($w = 1$) and *No E-Mail* as control ($w = 0$) with the *visit* label as outcome. With these treatment labels the dataset has 42,613 observations. We randomized the datasets and used 25% of each as test set in all experiments. The training set sizes vary in the experiments and are reported for each case separately.

For quantifying the accuracy of the uncertainty estimates we additionally created a semi-synthetic data that uses the covariates of `Starbucks` but generates outcomes using known functions. This is described in Section 4.3.4 together with the experiment.

**Model details.**  The DGP-models were trained with the RBF-kernel. We used the implementation provided by Gardner et al. (2018) following Milios et al. (2018), using gradient-descent for learning the prior noise and kernel length parameters to maximize the marginal likelihood. We used the Adam optimizer (Kingma & Ba, 2014) with learning rate 0.1 and a maximum of 1,000 iterations. The $\alpha_\epsilon$ was chosen based on log-likelihood of the training data amongst the set of $\alpha_\epsilon = 2^j$ for $j$ between $-1$ and $-7$ (i.e. 0.5 and 0.0078125).

For the honest tree we used half of the training to learn the tree, and the other half as the calibration set to estimate equation 8. We initially selected $M_{\max}$, the maximum number of leaf nodes, so that each leaf would contain on average 50 positive observations of the smaller of the two treatment groups for the full data. This resulted in 81 leaves for `Criteo`, 34 for `Hillstrom` and 12 for `Starbucks`. In addition, we required that each node contained at least $N_{\min} = 100$ observations in total. We will later study the effect of these parameters in Section 4.3.3. We based our implementation on Pedregosa et al. (2011).

We use the following metrics in our experiments:

**Area under the uplift curve (AUUC).** AUUC by Jaskowski & Jaroszewicz (2012) measures the expected increase in positive rate if targeting treatments with the model rather than randomly averaged over all possible treatment rates, and it is the standard metric for evaluating the overall goodness of uplift models. For a detailed explanation of the metric, see for instance Renaudin & Martin (2021). We report the results as units of 0.001 AUUC (mAUUC) for presentational convenience always computing AUUC for the test samples.

**Credible interval.** We characterise the uncertainty of the estimates via credible intervals, more specifically in terms of Highest Posterior Density-intervals (HPD-intervals) (Chen & Shao, 1999). We estimate these using Monte Carlo so that $S = 1000$ observations are drawn from $\mathrm{t}(\boldsymbol{x})$ and $\mathrm{c}(\boldsymbol{x})$ to obtain observations of $\mathrm{u}(\boldsymbol{x})$ using equation 2 and then we find the narrowest window containing a chosen fraction (we use 95%) of the observations. We call this the 95% *credible interval*.

**Average credible interval width.** We summarize the overall uncertainty of the whole data using the average of the 95% credible intervals over all *test set* observations, denoting this *the average credible interval width* (Average CI).

## 4.2 Illustration

Standard uplift models provide a narrow perspective to understanding the treatment effects. Here we demonstrate how uncertainty of the estimates can be used to improve the understanding of the data and to improve decisions.

Figure 1 (top) shows the uncertainty for an single observation in the `Starbucks` dataset for a DGP-model that was trained on 2000 observations. This user was chosen for having one of the highest uplifts among the observations and has a predicted uplift of 0.064, which is considerably higher than the average treatment effect of 0.0095. We observe that the uncertainty in estimating $\mathrm{t}(\boldsymbol{x})$ and $\mathrm{c}(\boldsymbol{x})$ is large. Consequently the distribution of $\mathrm{u}(\boldsymbol{x})$ is also fairly wide: the 95% credible interval is 0.257 and the probability of the treatment to have a negative effect is 0.168. Given access to this uncertainty, we can make rational decisions by e.g. maximizing expected utility. Figure 1 (bottom) shows a similar example for the honest tree trained on $400,000$ observations from the `Criteo` dataset. When training the model on more data the estimates have less uncertainty. However, this observation was selected for having relatively high uplift and happens to have large uncertainty.

Explicit quantification of the variation also allows improved investigation of the overall population, not just individuals. As an example, Figure 2 (top) provides a cross-plot of the expected uplift and the width of the 95% credible interval for the `Hillstrom` dataset. For this data, the variation is typically larger for the users with the largest expected effect which implies that we are actually quite unsure of the effect specifically for the users that would typically be targeted by the treatments. We also observe that the individual variation differs notably for cases with the same average effect, which allows more detailed identification of ideal candidates for the treatments. For example, for the case of $\tau(\boldsymbol{x}) = 0.1$ the 95% credible interval ranges from approximately 0.1 to 0.25, and only the individuals with the narrowest intervals could be treated without notable risk for negative effect. Without uncertainty estimates, this difference between the individuals would not be available. Figure 2 (bottom) shows a similar plot for the `Starbucks` data and we see that the two datasets are not alike; here the average 95% credible interval is largely independent of the average treatment effect and the average uncertainty is large for effectively all observations. In brief, $\mathrm{u}(\boldsymbol{x}) = 0$ is clearly within the 95% credible intervals for all instances and we cannot identify any individuals with reliable positive effect.

## 4.3 Evaluation

Having illustrated the possible use-cases for uncertainty estimates, we now proceed to quantify the behavior of uncertainty and the two proposed methods in more detail.

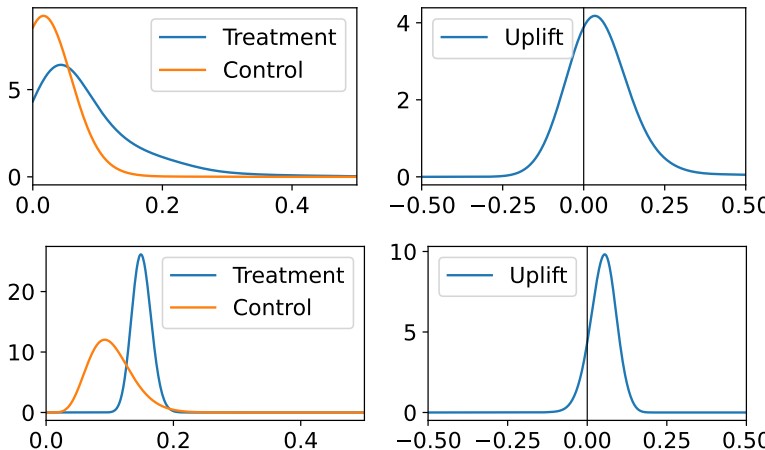

Figure 1: Uncertainty of response predictions (**left**) and uplift (**right**) of one test observation. **Top:** `Starbucks` with DGP trained on 2000 observations. **Bottom:** `Criteo` with tree trained on 400,000 observations.

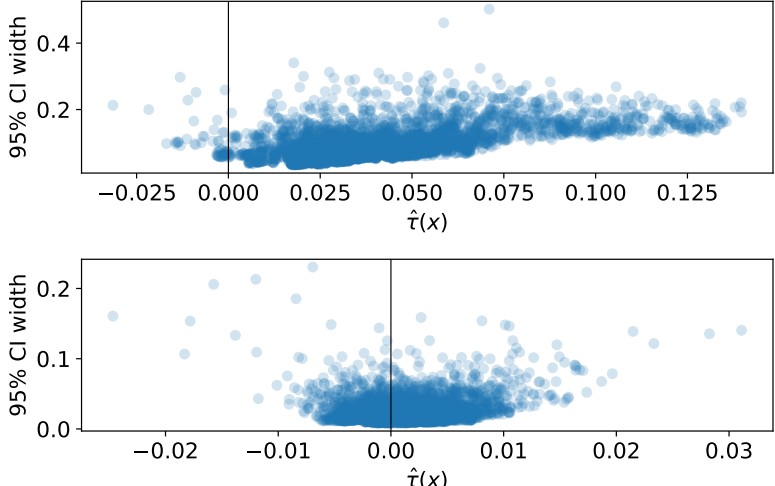

Figure 2: Width of 95% credible intervals vs. expected uplift. **Top:** DGP estimates for 32K samples of `Hillstrom` data. **Bottom:** DGP estimates for 32K samples of `Starbucks` data.

### 4.3.1 Amount of Data

Epistemic uncertainty should decrease when learning from larger datasets, and we start by empirically verifying this by training the models on subsets of varying size. Since we used exact inference for DGP we only run that for the smaller training sets (with at most 32k observations), whereas the fast tree is evaluated also for larger sets. The results are reported in Tables 1 and 2, and additionally visualized in terms of Average CI in Figure 3.

The main observation is that for both methods the Average CI reduces as a function of the available training data, confirming the expected behavior. The numerical values for both methods are similar but naturally not identical since the methods behave in rather different ways: The tree explicitly partitions the data into distinct leaves that all have identical uplift estimates, whereas the DGP fits a nonparametric estimate. For both methods the exact results depend on a few hyperparameters. For DGP we chose the parameter separately for each case based on log-likelihood, but for the tree we used the same parameters for every case

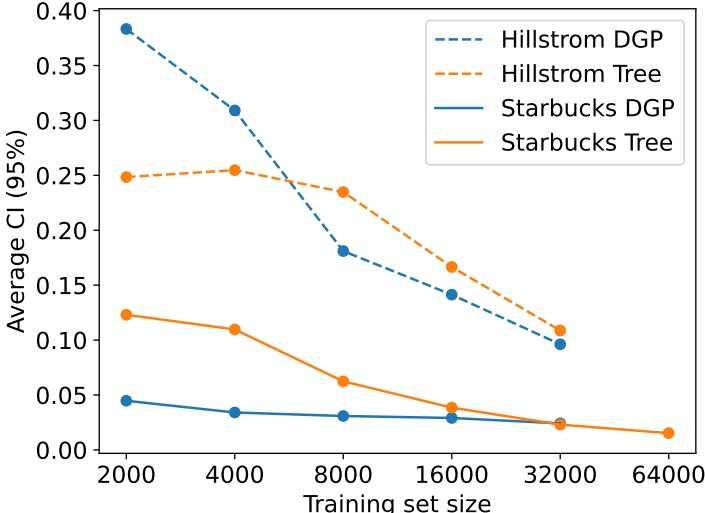

Figure 3: Average 95% credible interval width (Average CI) on the testing set, as a function of the amount of training data.

Table 1: Results for DGP for varying testing set sizes

| DATASET | SIZE | mAUUC | AVERAGE CI |
|---|---|---|---|
| Hillstrom | 2k | 0.318 | 0.383 |
| Hillstrom | 4k | -0.711 | 0.309 |
| Hillstrom | 8k | -0.001 | 0.181 |
| Hillstrom | 16k | 0.0862 | 0.141 |
| Hillstrom | 33k | 1.494 | 0.0962 |
| Starbucks | 2k | 0.683 | 0.0448 |
| Starbucks | 4k | 1.015 | 0.0341 |
| Starbucks | 8k | 1.526 | 0.0309 |
| Starbucks | 16k | 2.060 | 0.0291 |
| Starbucks | 32k | 2.533 | 0.0242 |

since the choice involves making a trade-off between accuracy of the mean estimates and the uncertainty (as will be shown in Section 4.3.3) and hence no obvious universal rule is available. This implies the choices for the tree are not optimal in terms of Average CI or AUUC, but a compromise between the two. One notable difference is also that the tree model uses only half of the available data for learning the tree structure and half for estimating the probabilities, and hence in practice has access to less data.

Another observation is that the AUUC metric is unstable. It should increase with the size of the training data until panning out for sufficiently large data, but especially for `Hillstrom` the values are essentially random. AUUC is known to be unstable for small data (Bokelmann & Lessmann, 2022) and previous authors have also observed that reliably estimating the uplift for this data is challenging (Nyberg & Klami, 2023; Rößler et al., 2021), but our results shed additional light on the reasons. AUUC depends on the ordering of the observations based on the mean estimates $\tau(\boldsymbol{x})$, but here the average uncertainty of these estimates is extremely large and any ordering is unreliable, as illustrated also in Figure 2. Consequently, an unstable AUUC is to be expected. For the other two datasets the AUUC behaves more like expected; for both `Starbucks` and `Criteo` the values for the smallest training sets are still noisy but we get consistent results on the larger training sets.

Table 2: Results for Honest Tree for varying testing set sizes

| DATASET | SIZE | mAUUC | AVERAGE CI |
|---------|------|-------|------------|
| Criteo | 100K | 0.378 | 0.00431 |
| Criteo | 200K | 0.173 | 0.00407 |
| Criteo | 400K | 0.420 | 0.00418 |
| Criteo | 800K | 0.341 | 0.00391 |
| Criteo | 1.6M | 0.397 | 0.00398 |
| Criteo | 3.2M | 0.369 | 0.00370 |
| Criteo | 6.4M | 0.319 | 0.00266 |
| Hillstrom | 2K | -0.384 | 0.248 |
| Hillstrom | 4K | -0.874 | 0.255 |
| Hillstrom | 8K | 3.358 | 0.235 |
| Hillstrom | 16K | 2.226 | 0.167 |
| Hillstrom | 32K | -0.662 | 0.109 |
| Starbucks | 2K | -0.324 | 0.123 |
| Starbucks | 4K | 0.0635 | 0.110 |
| Starbucks | 8K | 0.0610 | 0.0624 |
| Starbucks | 16K | 1.643 | 0.0385 |
| Starbucks | 32K | 2.079 | 0.0231 |
| Starbucks | 64K | 2.743 | 0.0153 |

Table 3: Results for comparison methods (above the line) and the proposed methods (below the line) on the Starbucks dataset with 32K observations in the training set. The comparison methods do not provide estimates of the confidence interval.

| MODEL | mAUUC | AVERAGE CI |
|-------|-------|------------|
| DC-LR | 2.332 | N/A |
| CVT-LR | 1.051 | N/A |
| Uplift RF | 2.823 | N/A |
| Honest Tree | 2.079 | 0.0231 |
| DGP | 2.533 | 0.0242 |

### 4.3.2 Comparison against baselines

Even though our main goal is in characterising the uncertainty of the estimates, it is important to validate that the newly proposed variants of double classifier and uplift trees are adequately accurate in the core task of estimating uplift. Table 3 compares the mAUUC against three benchmark methods representing examples of methods commonly used (Jaskowski & Jaroszewicz, 2012; Guelman et al., 2015; Nyberg et al., 2021; Nyberg & Klami, 2023; Semenova & Temirkaeva, 2019) for the `Starbucks` data trained on 32K observations. We use this data and train on large sample size to avoid possible issues with extreme class imbalance (of `Criteo`) or small sample size, and chose the baseline methods following Nyberg & Klami (2023).

Accounting for the natural variation in AUUC, all methods except CVT-LR are approximately in line. The best accuracy is obtained with DGP and Uplift RF, but we emphasize that the goal of this comparison is not to identify the best possible uplift method for this task but merely to illustrate that the proposed variants are reasonable. DGP is very similar to DC-LR that uses the same principle but different base classifier, and Honest tree is less accurate compared to Uplift RF likely because of using less data for training. When trained on 64K samples the honest tree uses approximately as many training instances for constructing the tree itself, and reaches mAUUC of 2.743.

### 4.3.3 Effect of Hyperparameters

In the previous experiment we used constant choices for the hyperparameters for the tree and selected hyperparameters based on log-likelihood for the DGP. Here we illustrate how these choices influence the results and provide suggestions on how they could be chosen in practice.

Figure 4 reports AUUC, Average CI, and the mean negative log-likelihood (MNLL) for u($\boldsymbol{x}$) averaged over the training samples for the DGP model as a function of its sole tuning parameter $\alpha_\epsilon$. Milios et al. (2018) suggested using the mean negative log-likelihood for selecting $\alpha_\epsilon$ for classification tasks and we see here that it is a reasonable criterion also for uplift. The choice of $\alpha_\epsilon = 2^{-5}$ that minimizes the MNLL also produces reasonable mAUUC, and the Average CI is in line with the metrics produced by the tree model. We chose to use the same $\alpha_\epsilon$ for both classifiers in order to avoid introducing two separate parameters, but it would be perfectly valid to use separate ones as well.

Figure 5 reports AUUC and Average CI for the tree model as a function of its two parameters $M_{\max}$ and $N_{\min}$. With the uncertainty based on the beta-difference distribution, less leaves (i.e. more observations in the leaves) generally leads to smaller Average CI, but increasing the number of leaves also enables the tree to potentially reach a higher AUUC before finally overfitting. Both of these trends are clearly present in the figure where the Average CI is narrowest when the tree size is heavily restricted by either parameter (bottom and left side of Average CI heatmap), and where the mAUUC is lowest both when the tree size is heavily restricted and when the three is not restricted by either parameter (top-right corner of mAUUC heatmap). The best mAUUC is found somewhere in the middle. In principle all of these trees are correct, but finding a *good* model is a trade-off between AUUC and Average CI. A practitioner should be aware of this compromise and we do not want to make a direct recommendation on what should be used as the exact criterion for making the choice, but note that selecting parameters that provide good AUUC would be a fairly natural choice.

An important observation is that the Average CI for the best parameters are essentially identical for both methods; 0.025 for DGP and 0.028 for the tree. Even though we do not have direct means of evaluating whether they are correct, the similarity of the two estimates obtained with very different methods is promising. Finally, Figure 6 shows the uplift curves for both methods using the optimal parameters, showing that both methods result in similar models.

### 4.3.4 Calibration of the uncertainty estimates

For the real data sets we cannot estimate whether the CIs are correct, and hence we use a semi-synthetic dataset to quantify the quality of the estimates. We re-use the seven-dimensional input features $\boldsymbol{x}$ of the `Starbucks` dataset to ensure the input distribution is realistic, and assume a simple non-linear mapping from those to the true probabilities:

$$P(\text{y} = 1|\boldsymbol{x}, \text{do}(w = 1)) = S(2x_1 + .3x_2 + .7x_3 - 1.5x_4 - .2x_5)$$
$$P(\text{y} = 1|\boldsymbol{x}, \text{do}(w = 0)) = S(2x_1 + .3x_2 + .7x_3 - 1.5x_4 - .2x_5 + 11.2x_6 \cdot 5.3x_7)$$

where $S()$ is the sigmoid function. We randomly assign the treatment label for every observation in the data to either $w_i = 1$ or $w_i = 0$. Based on this label, we sample $y_i$ from the Bernoulli distribution with corresponding $P(\text{y}|\boldsymbol{x}, \text{do}(w = w_i))$.

We interpret the uncertainty estimates from the perspective of CATE. For any given $\boldsymbol{x}_i$ the true ITE (Verbeke et al., 2022; Zhong et al., 2022) is

$$\theta_i = P(\text{y} = 1|\boldsymbol{x}_i, \text{do}(w = 1)) - P(\text{y} = 1|\boldsymbol{x}_i, \text{do}(w = 0)), \tag{9}$$

and for a local area of the input space (e.g. the contents of a leaf in a tree) we can compute the true CATE as the average

$$\bar{\theta}_i = \frac{\sum_{i \in l_m}(\theta_i)}{N_m} \tag{10}$$

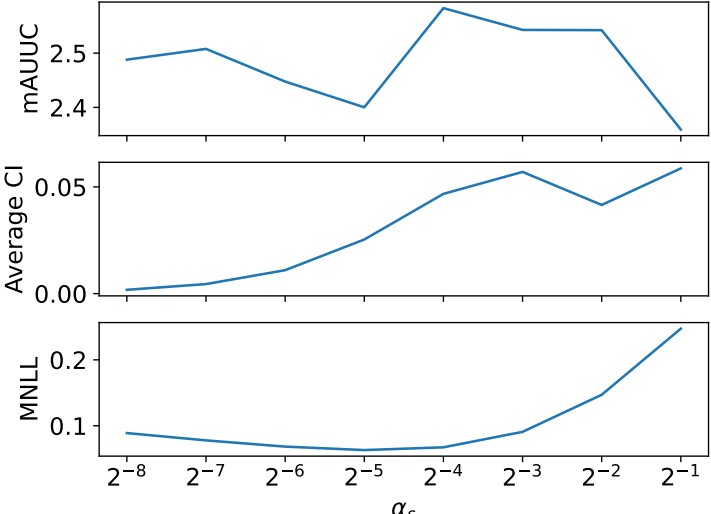

Figure 4: mAUUC, Average CI, and MNLL for DGP on `Starbucks` (32K) as function of $\alpha_\epsilon$. The optimal MNLL is at $\alpha_\epsilon = 2^{-5}$, resuting in Average CI of 0.0253 that is almost identical to the best tree model (see Fig. 5).

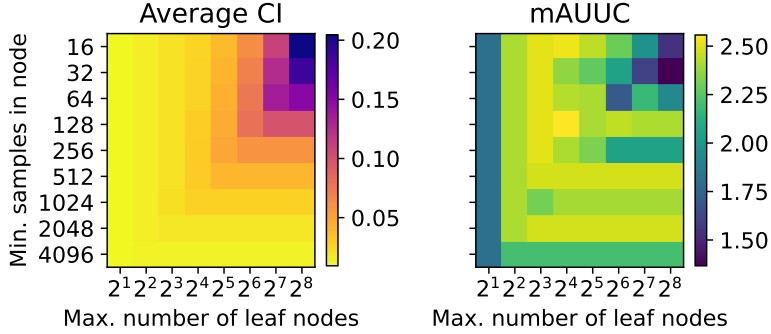

Figure 5: Average CI and AUUC for honest tree on `Starbucks` (32K) for different tree parameters. With $M_{\max} = 16$ and $N_{\min} = 128$ we get highest mAUUC of 2.557 with Average CI of 0.0284.

where the sum is over all observations $i$ in that area and $N_m$ is the number of observations in that area. For the Honest tree we evaluate the coverage of the CI with respect to $\bar{\theta}_i$.

Figure 7 illustrates the results for the Honest tree on this data, again covering the same range of hyperparameter choices as in the previous subsection to illustrate how the coverage varies. We also report the AUUC and Average CI for this specific data for completeness. The method is designed to estimate the uncertainty of CATE, and we see that the true CATE indeed falls typically inside the 95% CI and the results are consistent across a broad range of hyperparameters. While there is some variation in the models, 92% of the generating parameters $\bar{\theta}_i$ fell within the predicted 95% CI. Shuffling the data and re-training a model 288 times with max. leafs set to 12 and minimum leaf size set to 100, 95.0% of the generating parameters were within the predicted 95% CI's.

As the DGP does not have well defined local areas, we are unable to calculate the exact true CATE and cannot measure the real coverage even on simulated data.

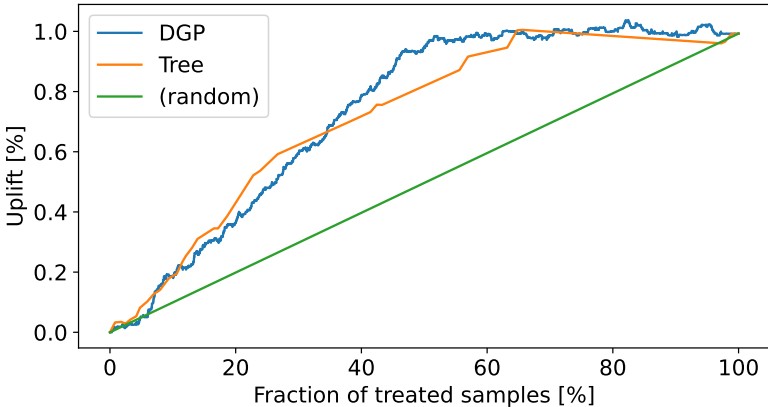

Figure 6: Uplift curve for Tree and DGP on `Starbucks` (32K) with optimal parameters, with the baseline of random targeting of treatments. For the tree the piecewise linearity is due to all samples in a leaf sharing the same estimate.

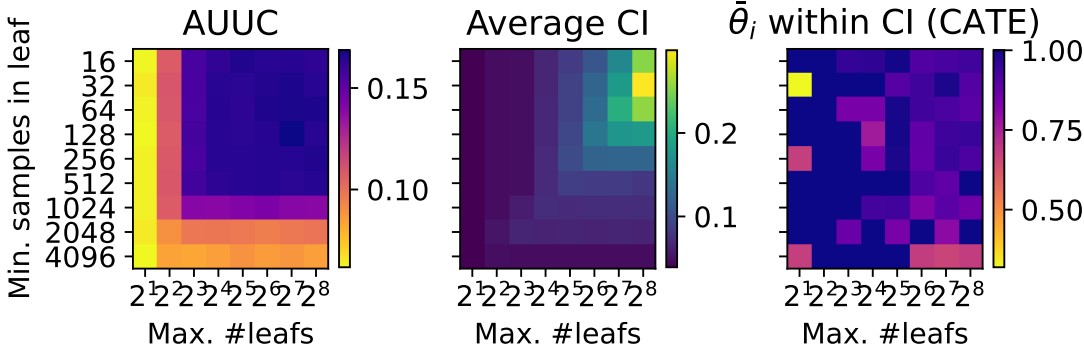

Figure 7: AUUC, average CI, and fraction of generating parameters $\bar{\theta}_i$ within estimated CI's for honest tree on semi-simulated Starbucks data (32K).

## 5 Discussion

Despite motivating the work in part by the need of quantifying the uncertainty in the medical domain, we only evaluated the method on e-commerce datasets due to lack of public medical data. This is understandable due to the sensitivity of medical data, e.g. as in the case of the data used by Kuusisto et al. (2014). For the `Hillstrom` dataset we confirm the earlier finding of high variability of the AUUC metric (Rößler et al., 2021). Previously this has been attributed to small dataset size, but our results reveal that the prime reason may actually be high variance of the estimates; the data is simply not informative enough of the potential effects.

Our experiments showed that the specific uplift models presented here solve the standard uplift estimation task with accuracy in line with the closest competitors, but extend their capability by providing the uncertainty estimates. On the real data sets we cannot make conclusions regarding the accuracy of the uncertainty estimates, and hence our main experimentation characterised how they depend on the amount of data and model hyperparameters. On synthetic data we showed that the CI estimates for the tree are well calibrated in terms of the true CATE, but for DGP we were unable to evaluate this since the true CATE is not well-defined. Overall, the empirical evidence regarding the accuracy of the estimates still remains partial, but we believe the experiments are enough to show that the models are useful and encourage researchers to account for uncertainty in their work and to evaluate the principle more thoroughly in future work. Empirical com-

parison against the estimates provided by the methods of Kallus et al. (2019); Jesson et al. (2021) would be valuable, but would need careful design due to different scopes; their key contributions are in addressing latent confounders that are not considered here, and the results would hence depend significantly on how the synthetic data is created.

The proposed uplift models follow the standard principles in the field, but the detailed methods are novel. We are not aware of double classifiers using the GP model by Milios et al. (2018) as base learners, or an honest tree that directly incorporates class imbalance correction. As shown in Table 3, for the mean uplift estimation task they are comparable to commonly used methods, confirming that the methods work well as uplift models. We presented these extensions because they include elements we consider worth mentioning, but remind that the general principle of estimating uncertainty described in Section 3 extends to a much broader class of models; for instance, all T-learners with uncertainty-quantifying base learners would work.

An interesting new use-case enabled by our approach is comparison of multiple treatments in scenarios where $\mathrm{u}_a(\boldsymbol{x})$ and $\mathrm{u}_b(\boldsymbol{x})$ have already been estimated for two treatments and we no longer have access to the original data. Since we now have distributions, we can still evaluate e.g. the probability $p(\mathrm{u}_a(\boldsymbol{x}) > \mathrm{u}_b(\boldsymbol{x}))$ to identify the preferred treatment for each individual.

## 6 Conclusion

We argued that quantifying uncertainty is important when estimating the individual treatment effects, both because of limited data for estimating the effect for each individual and because the methods are often used to make decisions that have significant effect on individuals. Even though the lack of methods quantifying uncertainty has recently been pointed out by other authors as well and there are some examples of models that provide uncertainty estimates presented in context of latent confounders, we still lack easy ways of accounting for uncertainty for the commonly used uplift modeling techniques. Our main goal was to raise awareness of this and to provide both a theoretical basis and easy practical methods for estimating the epistemic uncertainty of the estimates.

For the double classifier the estimates are well-calibrated when using well-calibrated classifiers due to the uplift corresponding directly to their difference, and for the tree models we get the exact distribution conditional on the finite sample in each leaf. Consequently, the methods presented here provide mathematically consistent estimates conditional on the modeling assumptions. However, in practice the estimates naturally depend on the practical details such as the hyperparameters used when training the models and the quality of the features characterising the individuals, and both models make independence assumptions that are likely not accurate in real scenarios. We showed that both approaches provide relatively consistent estimates of the uncertainty, but were only able to directly measure the calibration on artificial data and for the tree model. We see no clear reason to recommend either of the methods specifically, but rather both have advantages and disadvantages and for a practical task it is ideal to try out both.

Numerical evaluation of the reliability of the estimates remains as the most important future direction, but there is no reason to believe that the results shown here would not be qualitatively accurate. In other words, the main empirical result of the uncertainty being large is likely to hold and it itself is sufficient reason to pay attention to the uncertainty. For instance, the observation that the credible intervals for individual estimates are wide even for the large `Criteo` data is something every practitioner should be aware of and hence we would recommend e.g. practitioners in e-commerce and advertising to seriously consider uncertainty in their tasks using it e.g. to improve ad campaign reliability. In general, however, we see the highest value for these tools in applications where the treatments have significant personal effects, for instance in medical domains, personalised educational interventions, or career development support. We feel that in such applications it is crucial that future works always explicitly address the uncertainty in some way.

### 6.0.1 Broader Impact Statement

Uplift models are used to influence decisions at the level of the individual, and hence considerable care is needed when using them in any context. The fairness and ethical aspects of uplift modeling applications is determined by who's interest is optimized, and their relative weighting determines who ends up carrying the

risk and who ends up receiving the benefits. In some cases, it is the interest of the individual being targeted that is optimized, sometimes it might be the interest of the one targeting treatments. Our goal was to improve transparency of such decision-making by providing tools that allow characterising and communicating the reliability of the results, for instance to ensure that the potential gains and risks are rationally accounted for instead of unintentionally making decisions that may result in unnecessary harm. We think that it is important that this research is done in public and we also provide open source program code for reproducing our results.

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
