# OpenReview forum: "Quantifying Uncertainty of Uplift"
_TMLR — Rejected by TMLR_

### Review · Reviewer_uMFt · 2023-06-12

**Summary Of Contributions:**

This paper fills a void in the study of uplift modelling, aka Conditional Average Treatment Effect (CATE), on the uncertainty of its estimation. It takes the Bayesian approach to two commonly used uplift model families, double-classifier and tree-based methods, augment their base models with Bayesian models, and then calculate the credible interval of the uplift estimate. It demonstrates the behavior on three commonly used benchmark datasets and discusses some insights obtained from the additional uncertainty estimation.

**Audience:**

Yes

**Broader Impact Concerns:**

Sufficiently discussed.

**Claims And Evidence:**

Yes

**Requested Changes:**

- Consider adding a dataset with (synthetic) ground truth to study the CI estimation error. It's hard to assess empirically whether the proposed method is an accurate estimate of the CI or not without ground truth.
- Some places require additional discussion / clarificaiton:
  - Section 2: "the equations are valid also without this assumption given that the requirements are satisfied in some other way." What are "some other way"?
  - Section 3.1 Eq 5-6 contradicting with "sampling from the log-normal marginals". Is there a missing "exp" in the equation?
  - Section 3.2 "Reliable estimates ... Following this idea, we use Nˆ estimated from a separate calibration set for computing equation 8." Can you comment whether using using a separate calibration set will provided unbiased estimate in this work? The statement like "likely to improve the reliability" is a bit vague.
  - Discussion "but the detailed methods are novel". Why don't you use the uncertainty estimation in Huang et al. (2022) but use the DGP model as a "novel" method?

**Strengths And Weaknesses:**

Strengths:
- This paper is clearly presented with adequate background introduction, methodology explanation, experiments and discussion.
- It starts with a good motivation that obtaining the uncertainty estimation of the uplift model is essential in the medical domain and also useful in advertising scenarios.
- The description of the method that converts standard base models to Bayesian models and obtains the uncertainty estimate is clear and the method is reasonable, though a bit trivial
- It has sufficient discussion on the weaknesses of this paper, including (1) the lack of evaluation in the medical domain due to lack of public datasets, (2) the proposed method following standard principles to obtain Bayesian estimate

Weaknesses:
- Limited methodology contribution. As discusses above, the introduced Bayesian models are fairly straightforward on top of existing base models. In the introduction, the authors mention "Huang et al. (2022) ... settled for quantifying the uncertainties of the two classifiers in the double classifier, rather than the uplift itself." The additional methodology contribution of this paper in the two-classifier family appears to be estimating the CI of the difference of two classifier outputs using Monte Carlo? In the tree-based model family, interpreting the count based uplift estimate (Eq 7) using Beta-Bernoulli distribution is also commonly used in the Bayesian literature.
- Lack of evaluation of the estimated CI with ground-truth. As admitted by the authors, the three datasets do not contain ground truth to verify whether is estimated CI is accurate. Nevertheless, experiments do show the behavior how the estimated CI change with amount of training data and the effect of hyperparameters. It would still be valuable to construct a synthetic data / semi-synthetic model based on real data to study whether the estimated CI is accurate. It would also be useful for any follow up work to compare with this paper.

---

> ### Author Response · Authors · 2023-07-06
>
> Thank you for the detailed comments. We now uploaded a new version of the paper that addresses all major comments of all reviewers, and below we provide detailed answers to your remarks.
>
> > Limited methodology contribution. As discusses above, the introduced Bayesian models are fairly straightforward on top of existing base models. In the introduction, the authors mention "Huang et al. (2022) ... settled for quantifying the uncertainties of the two classifiers in the double classifier, rather than the uplift itself." The additional methodology contribution of this paper in the two-classifier family appears to be estimating the CI of the difference of two classifier outputs using Monte Carlo? In the tree-based model family, interpreting the count based uplift estimate (Eq 7) using Beta-Bernoulli distribution is also commonly used in the Bayesian literature.
>
> We believe methodological simplicity is a virtue in a paper that primarily aims at raising awareness of a neglected perspective, and our aim was specifically to introduce a basic principle that would be easy to build on. The technical elements are indeed fairly straightforward and we believe many researches would converge on similar choices (Monte Carlo estimate and Beta-Bernoulli), which should make it more likely for broader audience to be able to follow our main arguments. We hope that this also helps readers to trust our main empirical observation that the uncertainty of the estimates for common uplift benchmarks is large even when training on relatively large sample sizes.
>
> The revised Introduction now makes this perspective more clear.
>
> > Lack of evaluation of the estimated CI with ground-truth. As admitted by the authors, the three datasets do not contain ground truth to verify whether is estimated CI is accurate. Nevertheless, experiments do show the behavior how the estimated CI change with amount of training data and the effect of hyperparameters. It would still be valuable to construct a synthetic data / semi-synthetic model based on real data to study whether the estimated CI is accurate. It would also be useful for any follow up work to compare with this paper.
>
> Agreed. We now added a new experiment that directly quantifies the accuracy of the estimates, using a semi-synthetic data that uses real covariates of the Starbucks but known functions for the conversion rates. The uncertainty estimates are found to be fairly well calibrated for the tree for broad range of hyperparameters, but need to be interpreted with care as explained in Section 4.3.4.
>
> > Section 2: "the equations are valid also without this assumption given that the requirements are satisfied in some other way." What are "some other way"?
>
> One way to deal with this is to use inverse probability of treatment weighting (IPTW). We now mention this in the paper.
>
> > Section 3.1 Eq 5-6 contradicting with "sampling from the log-normal marginals". Is there a missing "exp" in the equation?
>
> Eq. (5) and (6) are correct but the writing was a bit confusing and is now fixed. Here $\hat y$ is log-transformed $\alpha$ that follows normal distribution, corresponding to $\alpha$ being log-normal.
>
> > Section 3.2 "Reliable estimates ... Following this idea, we use Nˆ estimated from a separate calibration set for computing equation 8." Can you comment whether using using a separate calibration set will provided unbiased estimate in this work? The statement like "likely to improve the reliability" is a bit vague.
>
> We used separate validation set based on the argumentation and empirical results of Athey and Imbens (2016) but did not directly validate the improvement. We now clarified the writing to better communicate this.
>
> Both training and validation data give unbiased estimates, but using a validation data reduces the risk of overfitting in cases where the sample count in either group is very small. We incorporated it into our model because of the results of Nyberg and Klami (2023) relating to the imbalance in the Criteo data we used, but the proposed method could just as well use estimate the rates from the training data itself.
>
> > Discussion "but the detailed methods are novel". Why don't you use the uncertainty estimation in Huang et al. (2022) but use the DGP model as a "novel" method?
>
> Huang et al. (2022) considered real-valued outcomes and hence their specific model could not be used here, but the point is still valid. We chose DGP because of nice computational properties and previous results showing good calibration, but we could indeed replace the DGP with any other GP-based model (or more generally any classifier that provides reasonable estimates of uncertainty).

---

### Review · Reviewer_6LAS · 2023-06-16

**Summary Of Contributions:**

The authors propose 2 strategies to provide uncertainty estimates on a treatment effect.
1. a double classifier approach using GPs with Dirichlet priors
2. Fitting beta distributions on uplift-trees' leafs

They compare these methods in an extensive experimental setup.

**Audience:**

Yes

**Broader Impact Concerns:**

n.a.

**Claims And Evidence:**

No

**Requested Changes:**

- I believe the notation is best to be overhauled to the literature standard (capitalised random variables), while also explaining the necessity of eq3 and eq4.

- CATE (and uplift modelling) is a well studied area, much more benchmarks need to be included. In fact the authors mention some benchmarks in their introduction, some of them using GPs and RFs (or single trees), these need to be included.


**Strengths And Weaknesses:**




WEAKNESSES
- I don't really like the notation. As an example, what is the reason for introducing a split equation in eq. 3? It seems each sub-equation is the same, besides the range of the integral, which simply follow each other? (Furthermore, it seems there is a bracket too many in the second sub-equation in eq.3)

- I also don't agree that t and c are independent (I am assuming here that the text below eq.4 contains a typo where you mean that you need to estimate p(t, c) to estimate p(u), given eq3 and eq4 above). Essentially, t and c should be confounded by X (making them dependent). This is one of the basic assumptions in CATE.

- Based on the rather strong assumption of p(t, c) = p(t)p(c) (which I criticise above), you are estimating each distribution separately. This actually corresponds with a simple T-learner. Note further that a T-learning makes no assumption on the used underlying model (be it a random forest, neural network, or indeed a GP with Dirichlet prior). It seems to me the authors are proposing a more limited version of a T-learner which is known in the literature since quite some time?

- The second method the authors propose quite unclear to me. Why are you using uplift trees? What's different between these trees and a standard decision tree? Are you fitting Beta distributions on the samples collected under one leaf?

- In your experiments, you are only comparing the proposals in this paper? I see no other benchmarks, am I missing something?


STRENGHTS
- The authors are correct, this is an interesting topic; uncertainty will absolutely guide to much more robust and informed decision making.

---

> ### Author Response · Authors · 2023-07-06
> **Included experiment on semi-synthetic data to verify that the predicted CI's are indeed valid**
>
> Thank you for the feedback. We now uploaded a modified version of the paper that accounts for comments from all reviewers, and below we respond to the specific remarks. We start with the requested changes and cover the other points after that.
>
> > I believe the notation is best to be overhauled to the literature standard (capitalised random variables), while also explaining the necessity of eq3 and eq4.
>
> The mathematical notation follows the recommendations of TMLR. We can easily change the notation if you insist, but for now we kept the journal recommendation.
>
> The equations 3 and 4 are in a sense obvious, but we feel that it is important to provide formal definition for the main quantities of interest.
>
> > CATE (and uplift modelling) is a well studied area, much more benchmarks need to be included. In fact the authors mention some benchmarks in their introduction, some of them using GPs and RFs (or single trees), these need to be included.
>
> It is important to make a distinction between methods that provide mean estimates of CATE and methods that provide also uncertainty of that. There is indeed vast literature on the former, but considerably less on the latter.
>
> As you pointed out, for the conventional task our methods are minor minor variants of standard approaches (T-learner and decision trees) and that was intentional. As we now clarify in the Introduction, providing easy ways of adding uncertainty quantification for the commonly used models was our main motivation.
>
> We previously verified that the methods work in the mean estimation task and mentioned this in the Discussion, but now added the results to the paper. Table 3 compares the methods against a range of benchmarks in terms of AUUC. The list of comparison methods does not even attempt to be exhaustive, since the result is provided merely as a verification of general behavior.
>
> We now added an experiment (Section 4.3.4) that directly measures the quality of the uncertainty estimates for semi-synthetic data. This makes the empirical validation stronger in general, despite not directly addressing the question of comparison methods.
>
> > don't really like the notation. As an example, what is the reason for introducing a split equation in eq. 3? It seems each sub-equation is the same, besides the range of the integral, which simply follow each other? (Furthermore, it seems there is a bracket too many in the second sub-equation in eq.3)
>
> These was indeed a minor typo, and as explained above we prefer to use notation recommended by TMLR.
>
> > I also don't agree that t and c are independent (I am assuming here that the text below eq.4 contains a typo where you mean that you need to estimate p(t, c) to estimate p(u), given eq3 and eq4 above). Essentially, t and c should be confounded by X (making them dependent). This is one of the basic assumptions in CATE.
>
> You are correct that they are not independent in general, and we now clarified the writing to better communicate that we only assume conditional independence given $x$, matching the basic assumption of many CATE models.
>
> > Based on the rather strong assumption of p(t, c) = p(t)p(c) (which I criticise above), you are estimating each distribution separately. This actually corresponds with a simple T-learner. Note further that a T-learning makes no assumption on the used underlying model (be it a random forest, neural network, or indeed a GP with Dirichlet prior). It seems to me the authors are proposing a more limited version of a T-learner which is known in the literature since quite some time?
>
> The method is indeed a special instance of T-learner as explained in the beginning of Section 3. The whole idea is that we present a general definition of how the whole family of T-learners can be extended to account for uncertainty. The revised version makes this argument more clear.
>
> > The second method the authors propose quite unclear to me. Why are you using uplift trees? What's different between these trees and a standard decision tree? Are you fitting Beta distributions on the samples collected under one leaf?
>
> You are correct in assuming that the tree is a regular decision tree, but the adjustments we have made to it are only applicable to uplift modeling. We indeed fit Beta distributions on the samples of each leaf.
>
> > In your experiments, you are only comparing the proposals in this paper? I see no other benchmarks, am I missing something?
>
> We now added comparison against classical CATE models in terms of AUUC and a new artificial data experiment to directly quantify the estimates, but cannot provide empirical comparison against previous methods in the latter task. We found empirical experimentation against the methods pointed out by Reviewer SAhj both infeasible in practice and slightly unfair for them as they are not directly intended for this specific task but rather solve a more general problem and define the uncertainty in a different manner.

---

### Review · Reviewer_SAhj · 2023-06-22

**Summary Of Contributions:**

The paper proposes two Bayesian models for quantifying uncertainty in conditional average treatment effects. The proposed approach is to estimate outcome regression functions in each arm of a trial, and to compute a posterior over their differences at each covariate value x. The behavior of these approach is demonstrated on several datasets, although no comparison to other methods is attempted.

**Audience:**

Yes

**Broader Impact Concerns:**

No specific concerns about broader impact.

**Claims And Evidence:**

No

**Requested Changes:**

Corresponding to each of the weaknesses above:

 1. More extensive treatment of previous work, and contextualization of the current work in that literature.
 1. Better discussion of the key assumption that t(x) and u(x) are independent. Alternatively, a modification to the method that accounts for uncertainty about this correlation, e.g., by integrating over different possibilities.
 1. Better discussion of when the proposed methods would be appropriate in practice.
 1. Comparison to previous methods, at least ones that attempt to quantify Bayesian uncertainty.
 1. Evaluations that provide some notion of fidelity to ground truth.

**Strengths And Weaknesses:**

S1: The paper motivates and sets up the uplift modeling problem in randomized experiments well.

S2: Methods are laid out clearly.

W1: The paper could do much more to review the literature on uncertainty quantification in CATE estimation, and to contextualize the work here in that literature. Much of this work goes a step further and also considers unobserved confounding, although the case in this paper can be treated as a special case:
 - http://proceedings.mlr.press/v89/kallus19a/kallus19a.pdf
 - https://projecteuclid.org/journals/annals-of-statistics/volume-50/issue-5/Bounds-on-the-conditional-and-average-treatment-effect-with-unobserved/10.1214/22-AOS2195.short
 - http://proceedings.mlr.press/v139/jesson21a/jesson21a.pdf
 - See also the extensive subliterature using BART for causal inference. E.g., citations in this review: https://www.annualreviews.org/doi/abs/10.1146/annurev-statistics-031219-041110, or this paper: https://arxiv.org/abs/1706.09523. These authors and their students have applied these methods in a wide variety of applications.
 - See also work on conformal inference for ITE's: https://academic.oup.com/jrsssb/article/83/5/911/7056131

W2: The estimated densities of treatment effects depend on the joint density of (t(x), u(x)), but the authors claim that these can reasonably be assumed to be independent. This assumption requires justification, and I do not think that it would be reasonable in general. Note that it is well-known that the quanties of ITE's are known to be unidentifiable because we cannot identify the correlation between potential outcomes within units. Thus, the method would need to account for this, rather than assuming the problem away. See, e.g., https://academic.oup.com/jrsssb/article/83/5/911/7056131 for discussion.

W3: The paper provides no discussion of when either of the proposed methods might be more appropriate or useful.

W4: Given the existing work on uncertainty quantification in CATE estimation, the lack of comparison to other baselines is a major weakness.

W5: The current evaluations provide no sense if whether the proposed uncertainty estimates are valid in any sense. It might make sense to at least provide a simulation study, or to evaluate the method on data simulated from other DGP's, such as data from previous ACIC data analysis challenges.

---

> ### Author Response · Authors · 2023-07-06
> **We now discuss the literature in more detail and added an experiment to verify the validity of the credible intervals**
>
> Thank you for the feedback. We now uploaded a modified version of the paper that accounts for comments from all reviewers, and below we respond to your specific remarks, quoting the main weaknesses as the context.
>
> > W1: The paper could do much more to review the literature on uncertainty quantification in CATE estimation, and to contextualize the work here in that literature. Much of this work goes a step further and also considers unobserved confounding, although the case in this paper can be treated as a special case:
>
> Thank you for the pointers. We agree that these works are highly relevant and we now discuss them in detail in the Introduction that has been largely re-written for the updated version. We were aware of the general research direction, but had not previously realised that some of those papers indeed explicitly discuss uncertainty intervals.
>
> In light of this, we modified our claims to acknowledge that recently authors have started to pay attention to uncertainty but the work remains in context of rather specialized solutions in works that focus on accounting for unobserved confounding but that we are missing tools that would be easy to integrate into the more straightforward models that are routinely used e.g. by the e-commercse industry. We now also emphasize our other key contribution more: Our empirical experiments show that for common uplift tasks and datasets the uncertainty of the estimates is extremely large, likely considerably larger than what most practitioners would intuitively think. This observation was initially our main motivation to write about the topic.
>
> > W2: The estimated densities of treatment effects depend on the joint density of $(t(x), u(x))$, but the authors claim that these can reasonably be assumed to be independent. This assumption requires justification, and I do not think that it would be reasonable in general.
>
> We clarify that we do not assume $t(x)$ and $u(x)$ to be independent in general, only conditional on $x$. This is the same assumption many of CATE models rely on. For smooth functions it holds for models using sufficiently local information for constructing the etimates (GPs with short lengthscales, trees with small leaves) but indeed does not hold in general.
>
> We now clarified both the text and notation regarding this assumption and extended the discussion.
>
> > W3: The paper provides no discussion of when either of the proposed methods might be more appropriate or useful.
>
> We intentionally did not make strong conclusions due to lack of clear evidence. We now revised the conclusions to clarify this.
>
> > W4: Given the existing work on uncertainty quantification in CATE estimation, the lack of comparison to other baselines is a major weakness.
>
> We could not find a realistic way to explicitly compare against the related work presented in the context of unobserved confounding, in part due to lack of open implementations and in part because the methods are primarily intended for use in a different scenario as we are not explicitly considering latent confounders. The paper hence remains suboptimal in this respect, but we made sure the claims we make regarding the algorithms take this into account and direct comparison against special cases of the previous approaches is listed as future work.
>
> We also note that the updated paper shows the new models are comparable in mean accuracy for a range of comparison methods practitioners are using. While this does not directly compare the uncertainties against other baselines, it should still help readers position the work within the broader literature.
>
> > W5: The current evaluations provide no sense if whether the proposed uncertainty estimates are valid in any sense. It might make sense to at least provide a simulation study, or to evaluate the method on data simulated from other DGP's, such as data from previous ACIC data analysis challenges.
>
> The updated version includes a simulated study that directly quantifies the accuracy of the estimates, as well as discussion on how to interpret the uncertainty intervals.

---

### Decision · Action_Editors · 2023-08-08

**Recommendation:** Reject

**Comment:**

Although the authors generally agree that the topic of this manuscript is of interest and that the contributions of the authors may have some merit, two of three reviewers voiced their concerns regarding the coverage of related literature on CATE estimation and uncertainty quantification. Further, the reviewers expressed concern regarding details of the assumptions made, the general approach taken, and the potential applicability of the proposed methods in practice.

These are serious concerns that would need to be overcome before this work could be accepted for publication.

---------- comments from reviewer recommendation added by EIC for clarity ------------------
I will reiterate my concern that the authors do not adequately characterize the literature on CATE estimation. In the statistical causality literature, variance estimation has always been a first-order concern, and the subtleties of estimating the variance of CATE vs ITE estimators (the first being identified and the second being unidentified) has been discussed at length. For example, the CATE variance estimation problem is so standard that in the Honest Trees paper, Athey and Imbens (2016) (https://www.pnas.org/doi/10.1073/pnas.1510489113#sec-5) simply state that: "Once constructed, the tree is a function of covariates, and if we use a distinct sample to conduct inference, then the problem reduces to that of estimating treatment effects in each member of a partition of the covariate space. For this problem, standard approaches are therefore valid for the estimates obtained via honest estimation and, in particular, no assumptions about model complexity are required." I had tried to provide examples in my first review that built on standard variance estimation results in causal inference. I would encourage the authors to reach further back in to the literature in statistics and econometrics, particularly in cases where parametric models are being used to estimate CATE; in most cases, these will look very much like quantifying the variance in each of the T-models (an approach that's cited in the introduction) because these models are estimated on independent data (which ends up mapping very closely to the methods the authors propose). I think the most appropriate framing of this work would be as an implementation of two specific Bayesian models.

I will also reiterate that point-estimating the joint density of (t(x), c(x)) is infeasible and is not necessary for estimating or quantifying the variance of CATE estimates. It is necessary for estimating the variance of ITE predictions. In particular, no assumption of independence is necessary to do CATE variance estimation because the object being estimated is the difference between two expectations (thus, quantifying uncertainty in each of the independently fit T-models is sufficient). However, it is unclear in the paper which of these two objects the authors are attempting to estimate, which I think has resulted in some confusion.

**Audience:**

Yes, the main focus of this manuscript -- causal inference and the estimation of conditional average treatment effects -- is of interest to a significant fraction to TMLR's audience.

**Claims And Evidence:**

The claims made in the manuscript, in its current state, are not completely supported by accurate, convincing, and clear evidence. I elaborate on this statement in the comments below.

**Resubmission Of Major Revision:**

The authors may consider submitting a major revision at a later time.